# Discrete Mechanistic Target of Rapamycin Signaling Pathways, Stem Cells, and Therapeutic Targets

**DOI:** 10.3390/cells13050409

**Published:** 2024-02-27

**Authors:** Meena Jhanwar-Uniyal, Sabrina L. Zeller, Eris Spirollari, Mohan Das, Simon J. Hanft, Chirag D. Gandhi

**Affiliations:** Department of Neurosurgery, Westchester Medical Center, New York Medical College, Valhalla, NY 10595, USA

**Keywords:** mTOR, mTORC1, mTORC2, S6K, 4E-BP1, GBM

## Abstract

The mechanistic target of rapamycin (mTOR) is a serine/threonine kinase that functions via its discrete binding partners to form two multiprotein complexes, mTOR complex 1 and 2 (mTORC1 and mTORC2). Rapamycin-sensitive mTORC1, which regulates protein synthesis and cell growth, is tightly controlled by PI3K/Akt and is nutrient-/growth factor-sensitive. In the brain, mTORC1 is also sensitive to neurotransmitter signaling. mTORC2, which is modulated by growth factor signaling, is associated with ribosomes and is insensitive to rapamycin. mTOR regulates stem cell and cancer stem cell characteristics. Aberrant Akt/mTOR activation is involved in multistep tumorigenesis in a variety of cancers, thereby suggesting that the inhibition of mTOR may have therapeutic potential. Rapamycin and its analogues, known as rapalogues, suppress mTOR activity through an allosteric mechanism that only suppresses mTORC1, albeit incompletely. ATP-catalytic binding site inhibitors are designed to inhibit both complexes. This review describes the regulation of mTOR and the targeting of its complexes in the treatment of cancers, such as glioblastoma, and their stem cells.

## 1. Introduction

Mechanistic target of rapamycin (mTOR; also known as mammalian target of rapamycin), an atypical serine/threonine (S/T) protein kinase, is a member of the phosphoinositide 3-kinase-related kinases (PI3K), which are conserved in all eukaryotes, localized in chromosome 1p36.22 [1]. The name mTOR originates from its inhibitor rapamycin, also known as sirolimus, which forms a complex with FK506-binding protein 12 (FKBP12) to inhibit its activity [2,3]. mTOR is a 289 kDa protein that regulates multiple cellular processes, including protein translation and metabolism. The deregulation of mTOR and associated proteins in its signaling pathway results in aberrant cellular growth, proliferation, migration, and survival, contributing to both the pathogenesis and therapy resistance of many cancers [2,3,4,5,6]. Structurally, the C-terminus contains a kinase domain, placing mTOR in the PI3K family, and includes the FKBP-rapamycin-binding (FRB) domain. Meanwhile the middle segment is categorized by a FAT (FRAP/ATM/TRRAP) domain, and the N-terminus mediates most interactions with the associated proteins via its numerous HEAT (Huntingtin; elongation factor 3, EF3; protein phosphatase 2A, PP2A; and kinase TOR1) repeats (See Figure 1A) [7,8].

mTOR functions by forming two major multiprotein complexes, mTOR complex 1 (mTORC1) and mTOR complex 2 (mTORC2) (See Figure 1B). mTORC1 is formed by the regulatory associated protein of mTOR (Raptor), proline-rich Akt substrate of 40 kDa (PRAS40), mammalian lethal with Sec-13 protein 8 (mLST8; also known as GßL), and DEP domain TOR-binding protein (Deptor) [3,9,10]. While Raptor does not possess any innate enzymatic activity, it is integral to the kinase activity of mTORC1 via its promotion of complex formation [11,12]. PRAS40 contains an mTOR signaling motif, and its overexpression competes with other mTORC1 targets for phosphorylation. PRAS40 responds to growth factor depletion to suppress mTORC1 activation [3,9]. Patients who underwent treatment with rapamycin had elevated activated PRAS40 expression and displayed levels of activated Akt as well as therapeutic resistance, suggesting the significance of activated PRAS40 as a surrogate marker for activated Akt [13]. Another component of mTORC1, mLST8, is limited in most mechanisms of mTORC1 activation but contributes to its activation by amino acids. While the upstream regulation of Deptor remains largely unknown, it is a component of mTORC1 and mTORC2, capable of inhibiting the activity of both complexes [14]. mTORC1 senses and controls cellular growth in response to various cellular signals, such as insulin or growth factors. Its targets have been previously characterized, particularly ribosomal S6 kinases and eukaryotic initiation factor 4E-binding proteins (eIF4E-BP1; also known as 4E-BP1). mTORC1 promotes the initiation of protein translation by associating with the eukaryotic initiation factor 3 (eIF3) complexes to phosphorylate these substrates. mTORC1 promotes protein translation via the activation of p70 S6 kinase (S6K) and the inhibition of 4E-BP1 and enhances RNA translation through the S6 ribosomal protein [2,15]. S6K exists as two distinct isoforms, S6K1 and S6K2; S6K1 consists of a 70 kDa cytoplasmic isoform and an 85 kDa nuclear isoform. Activated S6K appears to phosphorylate the 40S ribosomal subunit to increase the translational efficiency of a specific class of polypyrimidine mRNA transcripts, thereby regulating protein synthesis. S6K may also phosphorylate eukaryotic initiation factor 4B (eIF4B) at Ser422, allowing the association between eIF4B and eIF3, thus regulating translation via the promotion of eukaryotic initiation factor 4F (eIF4F) complex formation. Moreover, the phosphorylation of 4E-BP1 by mTOR regulates protein synthesis and promotes cap-dependent translation by releasing eIF4E, allowing its association with eukaryotic initiation factor 4G (eIF4G) among other factors. The translational control of nuclear S6K has been shown to regulate the transition from G1 to S phase in DNA synthesis [9].

In mTORC1’s downstream pathway, S6K performs various functions, including the phosphorylation of several targets, such as tumor suppressor programmed cell death protein 4 (PDCD4) or insulin receptor substrate-1 (IRS-1), thereby leading to its degradation and inhibiting PI3K and Akt in a known negative feedback loop [16,17]. Alternatively, Akt inhibition does not occur exclusively in the presence of insulin-like growth factor 1 (IGF-1); thus, mTOR may inhibit Akt via various mechanisms in the presence of different growth factors [18]. Recently, growth factor receptor-bound protein 10 (GRB10) has also been identified as an mTORC1 substrate [18,19,20]. mTORC1 simultaneously phosphorylates and stabilizes GRB10, further contributing to the feedback inhibition of the PI3K/Akt signaling pathway [21].

Notably, the mechanism by which rapamycin inhibits mTORC1, particularly mTORC1, activity is via the formation of a dimer with the small 12 kDa FKBP12, which then directly binds and potently inhibits mTOR [22,23]. Meanwhile, upstream of mTORC1 and mTORC2, Ras homolog enriched in brain (Rheb), a GTPase, also directly binds mTOR but results in the activation of mTORC1 [24]. This represents a critical mechanism by which extracellular and intracellular stimuli influence mTORC1 activity [25]. The tuberous sclerosis complex (TSC1/2), composed of TSC1 (hamartin) and TSC2 (tuberin), regulates Rheb1 activity via its GTPase-activating protein (GAP) activity [26,27]. The phosphorylation of TSC2 through the activation of the upstream PI3K/Akt and extracellular signal-regulated kinase (ERK1/2)-mitogen-activated protein kinase (MAPK) signaling pathways disables the GAP activity of TSC1/2, allowing for the disinhibition of Rheb1 and the activation of mTORC1 [2,28,29,30].

mTORC2 is composed of rapamycin-insensitive companion of mTOR (Rictor), stress-activated protein kinase interacting protein 1 (Sin1), and protein observed with Rictor (Protor) [10,31,32,33,34]. Structurally, mTORC2 is a homodimer of mTOR, Rictor, Sin1, and mLST8 heterotetramers [3,35]. Rictor and Raptor are mutually exclusive in their binding to mTOR. Notably, Rictor makes mTORC2 insensitive to rapamycin by masking the FRB domain of mTOR; therefore, FKBP12-rapamycin is unable to bind the Rictor-containing mTOR complex, and thus, it does not affect S6K [35]. The main function of Rictor is to provide scaffolding and regulate further substrate recruitment by mTORC2, while Sin1 determines the subcellular localization of mTORC2 and contributes to the substrate binding site [36,37]. Although the stability of the complex is dependent on other mTORC2 components, Rictor facilitates the binding of Protor 1 and 2 to mTORC2. Protor 1 and 2 are not involved in the catalytic function of mTORC2, but they play a significant role in the regulation of serum/glucocorticoid-regulated kinase 1 (SGK1) activity [38]. Sin1, another binding partner, is critical for the regulation of mTORC2 substrate specificity secondary to its promotion of Rictor–mTOR binding [38,39]. For example, the N-terminus of Sin1 is essential to aid in the binding of Rictor to mLST8 [35]. In addition, the Pleckstrin homology (PH)-domain of Sin1 has been shown to mediate the association of mTORC2 with cellular membranes [40]. The mutation of the Sin1 key domain has recently been implicated in cancer development due to its disruption of mTORC2 with resultant sustained Akt activation [41]. Furthermore, the kinase domain of mTOR controls the integrity of mTORC2 via the phosphorylation of Sin1, maintaining its protein stability and preventing its lysosomal degradation [42].

There are several overlaps in the mTORC1 and mTORC2 subunits, including mTOR, Deptor, and mLST8. Deptor aids in the regulation of these complexes by binding to mTOR via its PDZ domain to inhibit mTORC1 and mTORC2 [43]. On the other hand, mLST8 may play an important role in regulating the dynamic equilibrium between mTORC1 and mTORC2 in cells due to its integral role in maintaining mTORC2’s Rictor–mTOR interaction [36].

Among other members of the S/T protein kinase (SGK) family, Akt, also known as protein kinase B, is a major substrate of mTORC2 [10,31,44,45]. mTORC2 phosphorylates the C-terminal hydrophobic motif of these kinases in response to growth factor signaling. Akt was originally discovered as a proto-oncogene but has since demonstrated critical involvement in the regulation of various cellular functions, including transcription, protein synthesis, metabolism, cellular growth, proliferation, and survival. Akt lies in the interface between PI3K and mTOR, which is a part of the canonical PI3K/mTOR pathway. Although mutations in Akt are rare, ample mutations in upstream effectors of Akt, such as PTEN and PI3K, have been identified. mTORC2 directly phosphorylates Akt at the site responsible for maximal Akt activation, its hydrophobic motif Ser473 [46]. Akt is subsequently phosphorylated at Thr308 by phosphatidylinositol-dependent protein kinase 1 (PDK1), resulting in its full activation. The Akt phosphorylation of protein kinase C α (PKCα) regulates the actin cytoskeleton and various cellular functions; this pathway is crucial for the maintenance of normal and cancer cells through its involvement in multiple physiological functions, including cell cycle progression, transcription, translation, cellular differentiation, metabolism, motility, and apoptosis [12,31,34,45]. Additionally, upstream signaling by various stimuli, including growth factors and insulin, increases PI3K signaling, thus leading to the activation of mTORC2 by promoting its association with ribosomes; as a result, mTORC2 phosphorylates Akt at Thr450 in addition to its well-known site, Akt^Ser473^ [47]. This pathway, stimulated by PI3K signaling in cancer cells, can be disrupted by inducing apoptosis in phosphatase and tensin homolog (PTEN)-deficient cells [47]. The interaction between mTORC1 and mTORC2 is further illustrated as both complexes influence one another; Akt regulates PRAS40 phosphorylation, disinhibiting mTORC1 activity, and S6K regulates Sin1 to modulate mTORC2 activity. Recently, it has been suggested that mTORC2 activation can also occur via a PI3K-independent mechanism, involving the Yes-associated protein (YAP)/Hippo, Notch, and small G-protein Rac1 pathway [48].

mTOR is primarily localized in the cytoplasm; however, in response to certain growth factors, it can transport to the nucleus [33]. While its role in the nucleus is not fully understood, mTOR has been suggested to function via nucleocytoplasmic signaling [49]. Activated mTOR has been localized in subnuclear structures that resemble polymorphonuclear (PML) bodies, which are associated with Akt activation and control cell proliferation, apoptosis, and cellular senescence [50]. The presence of S6K has been shown in the nucleus as well as in cytoplasm [51]. Recent studies have shown that the nucleocytoplasmic shuttling of mTOR commonly occurs, and that phosphorylated S6K and Raptor play an important role in it [52]. In quiescent glioblastoma (GBM) cells, the platelet-derived growth factor (PDGF)-induced localization of mTOR in the nucleus was reduced by pretreatment with rapamycin [33]. 

Rapamycin and its analogues, termed rapalogues, act as allosteric inhibitors of mTOR; however, due to their incomplete mTORC1 inhibition or the loss of negative feedback loops resulting in unexpected mTORC2 activation, they have generally been ineffective in GBM clinical trials [53]. These types of inhibitors were called “first-generation mTOR inhibitors,” and their drawbacks led to the discovery of novel ATP-binding inhibitors of both complexes, which simultaneously suppress mTORC1 activity while effectively inhibiting mTORC2 activity, as demonstrated by the complete dephosphorylation of the mTORC1 downstream substrate pS6K^Ser235/236^ and the mTORC2 substrate pAKT^Ser473^ (See Figure 1B). These ATP-binding inhibitors were termed “second-generation mTOR inhibitors,” also known as “TORKinib” or “TORKi.” These concurrent mTORC1 and mTORC2 inhibitors have effectively targeted the proliferation and self-renewal of GBM cancer stem cells. Thus, the effectiveness of mTOR inhibitors in cancer therapy can be evaluated by their ability to suppress both complexes along with their degree of interference in both cellular proliferation and migration.

A “third generation” of mTOR inhibitors, namely, RapaLink-1, was invented to overcome resistance to rapalogues and TORKi. RapaLink-1 is a bivalent mTOR inhibitor formed by linking rapamycin with MLN0128, an ATP-binding inhibitor, thus consisting of an FRB domain linked to a TORKi. Consequently, with its targeting of both the FRB and the kinase domains of mTOR, RapaLink-1 has demonstrated the ability to target breast cancer cells with somatic mutations in mTOR, FRB, or the kinase domain, which typically confer drug resistance [54]. RapaLink-1 potently inhibits the mTORC1 pathway via its inhibition of 4E-BP1 phosphorylation, resulting in growth inhibition at levels comparable to rapamycin alone or in combination with MLN0128 both in vitro and in vivo.

## 2. mTOR Pathways in Cancer

mTOR has been shown to frequently undergo aberrant activation in cancer [3,55]. In fact, mTORC1 has been found to play a prominent role in the growth of established tumor cells [56]. The activation of mTOR is often the result of mutations in upstream regulators, such as a gain-of-function of PI3K or epidermal growth factor receptor (EGFR) and/or loss-of-function of tumor suppressor gene PTEN [57,58,59,60,61,62,63,64]. The activation of mTOR signaling drives the variations seen in cancer cell metabolism, including pathways for amino acid, glucose, nucleotide, fatty acid, and lipid metabolism [3,8]. In addition, other signaling molecules with oncogenic potential, such as RAS, can also stimulate mTOR signaling [65]. With its regulation of protein translation, ribosome biogenesis, cell proliferation, metabolism, and survival, mTOR has emerged as a promising chemotherapeutic target, used solely or combined with other chemotherapeutic agents [2,3,66,67]. 

As previously described, rapamycin and rapalogues act as the partial inhibitors of downstream effectors in the mTORC1 pathway, mainly 4E-BP1, while concurrently causing a compensatory increase in mTORC2/Akt activity [33]. Interestingly, rapamycin can inhibit or activate mTORC2 in certain cancer cells; while the exact mechanism of the latter remains to be elucidated, rapamycin may inhibit mTORC2 by stalling its assembly [33,68]. The PI3K/Akt pathway has been implicated in pathophysiology and resistance to chemotherapy in many solid tumors [10,69]. Although the regulation of mTORC2 is only recently becoming more understood, mTORC1 is known to be activated by both extracellular as well as intracellular stimuli, including growth factors and amino acids [62]. As discussed previously, Rheb1 and its disinhibition by upstream Akt activation represent one such mechanism by which mTORC1 is activated via these signals. Moreover, a major obstacle in the targeting of mTORC1 is its sensitivity to nutrients, and variations in cancer cell metabolism may complicate mTORC1 suppression by contributing to a state of persistent activation. Nevertheless, in tumors with mTORC1 activation, such as subependymal giant cell astrocytomas (SEGA) of tuberous sclerosis (TS), rapalogues have shown significant efficacy. For example, everolimus has demonstrated a 75% response rate in SEGA [70]. Numerous early clinical trials in recent years have investigated the safety and efficacy of rapamycin, rapalogues, and ATP-binding mTOR inhibitors as monotherapy or in combination with other agents in the management of various malignancies are presented in Appendix A.

## 3. Targeting mTOR and PI3K in GBM

GBM, the most common primary brain tumor in adults, has an incidence of approximately 10,000 cases per year in the United States [71]. The Cancer Genome Atlas (TCGA) network classifies GBM into four molecular subtypes based on specific gene alterations: proneural, neural, classical, and mesenchymal transcriptomic [72,73,74]. The signal transduction cascade of EGFR is often altered in GBM, with the extensive genomic analyses of human GBM samples demonstrating the genetic mutations of EGFR in approximately 57% of GBM patients [72,75]. Aberrant signaling of mTOR is linked to tumorigenesis of numerous malignancies, including GBM. The mutation in tumor suppressor PTEN as well as loss of heterozygosity on chromosome 10q (LOH10q) occurs frequently in both primary as well as secondary GBM [76]. Also, mutations in PTEN were seen in up to 36–60% of GBM [72]. Aberrant EGFR signaling and the loss of PTEN both lead to the activation of the PI3K/Akt/mTOR pathway, thus suggesting a potential therapeutic advantage of inhibition of this pathway [72,77,78]. Further, genetic studies have identified the activation of receptor tyrosine kinase/PI3K in 86% of GBM samples [72]. This increased the activation of the Akt/mTOR pathway that stimulates cellular growth, proliferation, migration, and survival, which are major hallmarks of GBM cells [2,79]. It is possible that mTOR activation is the major cause of GBM’s relentless growth and dissemination [79,80]. Consequently, multiple clinical trials have been initiated to investigate the therapeutic response, toxicities, changes in metabolism and biomarkers, and predictors of response to PI3K/mTOR inhibitors in GBM. These trials have predominantly been conducted in recurrent GBM with a smaller subset of newly diagnosed GBM. Further, mTOR inhibitors were evaluated in a variety of combinations, such as monotherapy, in conjunction with standard of care therapy, and/or in combination with other pathway inhibitors. Table 1 represents the ongoing clinical trials using PI3K, Akt, and mTOR inhibitors in GBM. Here, we discuss some of these trials.

Multiple phase II clinical trials using rapalogues showed limited success in producing meaningful clinical results, attributable to the concurrent inhibition of negative feedback loops in addition to crosstalk with other mitogenic pathways (NCT00515086, NCT00016328, NCT00022724) [81]. As PI3K is negatively regulated by PTEN, one trial by Cloughesy et al. in 2008 treated 15 patients with PTEN-deficient recurrent GBM with one week of oral sirolimus daily prior to re-resection and continued postoperatively until progression [13]. Of the 14 tumor samples with adequate tissue, rapamycin was detected in all 14 during re-resection. While 7 of 14 patients demonstrated decreased tumor cell proliferation according to the Ki-67 proliferative index, the remaining patients were noted to have Akt activation with increased PRAS40 phosphorylation, thought to be secondary to the loss of the negative feedback loop. The compensatory Akt activation in this subset of patients was correlated with a shorter progression-free survival (PFS), suggesting the failure of sirolimus in the treatment of PTEN-deficient GBM.

Temsirolimus (CCI-779), another mTOR inhibitor, was trialed in 43 recurrent GBM patients at a weekly intravenous dose of 250 mg. This dose was tolerated without any serious toxicities; however, median time to progression was only 9 weeks, indicating limited efficacy as a monotherapy using temsirolimus [81]. A larger phase II trial of 65 recurrent GBM patients showed more promising results in a subset of patients with higher baseline tumor levels of phosphorylated p70S6K. These patients had a significantly longer PFS of 5.4 months compared to 1.9 months. Overall, 36% of patients demonstrated radiographic improvement, and there was a 51% incidence of grade 3–5 toxicities [82].

A phase I study was conducted on 10 recurrent GBM patients with four days of mTOR inhibitor ridaforolimus administered daily at an intravenous dose of 12.5 to 15 mg prior to re-resection and continued postoperatively [83]. Ridaforolimus demonstrated its ability to cross the blood–brain barrier and inhibit mTOR activity, as evidenced by a decrease in its downstream effectors in tumor specimens; pS6 levels were reduced, and phosphorylated 4E-BP1(p4E-BP1) was reduced by >80% compared to patient serum baseline [81]. A subsequent study demonstrated a dramatic down-regulation of pS6K with combined PI3K/mTOR inhibition [33]. These studies, while not clinically successful, laid the foundation for the possible utility of combination PI3K/mTOR therapy [13].

Several phase I and phase II trials have also evaluated the efficacy of mTOR inhibitors as an adjunct to the standard of care treatment for GBM. Standard temozolomide (TMZ) and radiation therapy (RT) were supplemented with mTOR inhibitor everolimus in a phase 1 trial of 18 newly diagnosed GBM patients. Over a median follow-up of 8.4 months, nine patients (50%) developed grade 3/4 toxicities. Stable disease occurred in 14 patients, and 4 patients had partial response, as shown by imaging analysis [84]. In an additional study, 100 newly diagnosed GBM patients were administered everolimus for 1 week prior to conventional TMZ/RT and continued until progression. Overall survival for one year was seen in 64%, while median PFS was 6.4 months, demonstrating no appreciable survival benefit of standard therapy in conjunction with everolimus compared to controls [85]. Another phase 1 dose-escalation trial evaluating temsirolimus in 12 GBM patients in combination with TMZ/RT observed grade 4/5 infections in 25% of patients. Confining temsirolimus to the initial radiation phase instead of continuing it with TMZ adjuvant therapy and adding prophylactic antibiotics reduced the infection rate, though 2 of 13 patients in the second cohort exhibited worsening of pre-existing viral and fungal infections [86].

PI3K inhibitors may be isoform-selective or may target all four isoforms. Buparlisib (BKM120) is a pan-PI3K inhibitor, an ATP-competitive inhibitor targeting all isoforms, and it was evaluated in a phase I clinical trial in conjunction with standard therapy for newly diagnosed GBM patients, but the trial was discontinued due to significant toxicities [87]. In patients with recurrent GBM, phase II clinical trials demonstrated adequate brain tissue penetration of BKM120; however, it failed to render a significant clinical response as monotherapy or in conjunction with re-resection [58,87]. Another PI3K inhibitor is the combined PI3K/mTOR inhibitor, voxtalisib, which underwent a phase I trial in the treatment of high-grade gliomas. When administered in conjunction with TMZ with or without RT, stable disease was seen in 68% of patients, and partial response was achieved in 4% of patients. Lymphopenia (13%) and thrombocytopenia (9%) were the most frequent serious adverse events [88].

Despite advances in mTOR-targeting therapies, it is thought that the activation of mitogenic pathways and RAS/ERK1/2 via feedback loops contributes to the resistance of GBM [53]. The goal of active site inhibitors is to simultaneously target mTOR’s function in cell growth and proliferation along with its feedback loops. Second-generation mTOR inhibitors, termed “TORKinibs”, inhibit both mTORC1 and mTORC2 via allosteric interactions with the ATP-binding pocket [89,90]. Several small molecules have been identified that act as ATP-competitive inhibitors of mTOR, including PP242, KU0063794, AZD3147, and eCF309, among others. Pyrazolopyrimidines, PP242 and PP30, are potent mTOR inhibitors that display a high degree of selectivity towards mTOR relative to PI3Ks and other protein kinases. Meanwhile, KU0063794 also showed promise in suppressing cell cycle and proliferation compared to the selective PI3K inhibitor, LY294002, or combined PI3K/mTORC1 inhibitor, PI-103 [91]. 

Novel ATP-competitive mTOR inhibitor, Torin 1, of the quinoline class, inhibits both mTOR complexes [92]. Drawbacks of Torin 1 include its water insolubility and rapid metabolism by the liver that result in poor bioavailability and a relatively shorter half-life [93]. Consequently, the related compound Torin 2, which was created by Liu et al., exhibits improved water-solubility with increased oral bioavailability and a longer half-life [93]. Torin 2 has also emerged as a potent mTOR inhibitor capable of suppressing cellular proliferation and migration in GBM [94,95]. This recent study demonstrates that Torin 2 is the only mTOR inhibitor with the ability to suppress the self-renewal of cancer stem cells (CSCs) in GBM. Another recently discovered ATP-competitive mTOR inhibitor, XL388, of the benzoxazepine class, works similarly to Torin 1 and Torin 2. As with Torin 2, this drug demonstrates sufficient oral bioavailability and efficacy at low concentrations, as well as selectivity of mTOR over PI3K [96].

As opposed to mTORC1 inhibitors, the inhibition of Akt^Ser473^ phosphorylation by combined mTORC1 and mTORC2 inhibitors produces superior outcomes in GBM treatment. Compared to rapamycin, these novel inhibitors also demonstrate the superior suppression of p70S6K phosphorylation, and PP242 displays the enhanced suppression of GBM cell proliferation and migration [97,98]. Similarly, in vivo studies investigating the ATP-competitive mTOR inhibitor, AZD8055, demonstrate the reduced phosphorylation of both S6 and Akt with the subsequent reduction of tumor growth [99,100]. Clinical trials are now evaluating these dual inhibitors, including AZD8055 and sapanisertib (MLN0128, TAK228).

## 4. The mTOR Pathway and Stem Cells

mTOR functions via two distinct protein complexes, mTORC1 and mTORC2, which are a part of the key intracellular signaling pathway of PI3K/Akt/mTOR. These complexes respond to different signals, including growth factors and cellular energy to regulate cell growth and shape by influencing cytoskeletal remodeling, determining when and where cells grow as dictated by mTORC1 and mTORC2, respectively [8,101,102]. mTORC1 regulation of protein synthesis, via S6K and 4E-BP1, and initiation of protein translation have played a critical role in the maintenance of stem cells [103]. The mTOR pathway functions in normal neural stem cells (NSCs) by altering gene transcription for normal cell growth and migration processes [31]. Furthermore, studies have shown that mTORC1 is essential for the proliferation and survival of neural stem cells and appears to regulate their postnatal differentiation in the subventricular zone (SVZ) [104]. For example, insulin encourages neural differentiation via the PI3K/Akt/mTOR pathway by inducing mTOR phosphorylation [105,106,107]. Also, the activation of mTOR via suppression of its negative regulators, such as PTEN or TSC1, was shown to promote axonal regeneration [108]. These observations imply that alterations in the mTOR pathway can lead to severe deficits in nervous system development, resulting in various abnormalities, such as tumors, autism, and seizures [8]. Suppression of mTORC1 may inhibit NSC differentiation and decrease the population of neural progenitor cells. Alternatively, the hyperactivation of mTORC1 may also lead to the depletion of NSC progenitor cells due to accelerated differentiation [109,110]. Also, one observation revealed the gradual loss of NSCs in the neonatal SVZ which led to terminal differentiation and not proliferation after mTORC1 hyperactivation [111]. Furthermore, reduction in neural differentiation of NSCs following mTOR inhibition with rapamycin was seen without changes in proliferation [107]. Studies have shown that the mTOR pathway is also associated with dendrite formation and signal transduction between neurons, an additional function of mTOR besides neural regulation, as demonstrated in the development of newly born olfactory bulb neurons and their dendrites [104]. In addition, both mTOR complexes are shown to be involved in the dendritogenesis of SVZ-derived neurons [104]. These studies support the pivotal role of the mTOR pathway in neurogenesis via its influence on NSC function.

Recent studies have identified small subpopulations of cells present within the tumor and surrounding areas, termed CSCs, that are capable of self-renewal and play a critical role in tumor initiation, progression, maintenance, and recurrence. Additionally, these CSCs remain resistant to conventional chemotherapy and radiation therapy; therefore, they are the prime cause of therapy failure as well as cancer recurrence. Thus, understanding the origin and characteristics of CSCs will elucidate the mechanism that regulates stemness and drug resistance, leading to effective treatment of cancers, by targeting CSCs [112]. The existence of CSCs has been well established in brain tumors, such as GBM and medulloblastoma [112,113,114,115,116,117]. It is thought that the resilience of the CSC populations allows for the recurrence and invasion of GBM despite aggressive chemotherapy [114]. Particularly, studies have demonstrated CSCs in GBM can enter a quiescent state, rendering a refractory status with lower susceptibility to therapies [118]. Further, CSCs have been identified beyond the tumor margin of GBM; thus, the presence of these CSCs in the peritumoral area has emerged as an important prognostic marker, as measured by the expression of stem cell marker Nestin in relation to c-Jun N-terminal kinase (JNK)/MAPK [119].

GBM stem cells and CSCs are under the influence of several signaling pathways, namely, the PI3K/Akt/mTOR and EGFR pathways [120]. Nevertheless, the regulation of the molecular aspects of GBM stem cells remains elusive. Ample evidence demonstrated that the mTOR pathway regulates CSCs and has a significant role in the persistent growth and invasion of GBM stem cells. The deregulated mTOR pathway has been implicated in multiple cancer types, such as breast and renal cancer, and the inhibition of this pathway has shown therapeutic potential [3,10,121,122]. A study evaluating the interference of this pathway in CSC by rapamycin or rapalogues demonstrated that the proliferation of stem cells was significantly curtailed. Further, the Akt/mTOR signaling pathway has been implicated in maintaining the stem cell properties of CSCs in glioma [123]. Therefore, combined targeting of this pathway has been evaluated for the potential targeting of GBM stem cells and the improved treatment of GBM [124,125,126].

The role of mTOR in GBM stem cell regulation is just beginning to be achieved. However, there is evidence that the mTOR pathway can regulate hematopoietic stem cells, via its role in controlling autophagy [127]. Thus, a benefit of mTOR inhibition is the induction of autophagy in GBM stem cells with resultant antiproliferative and antidifferentiation effects [128]. Rapamycin and its analogues have been shown to incompletely inhibit mTORC1 from executing its pro-growth functions [7,23]. Therefore, approaches to inhibit multiple deregulated pathways by targeting both mTORC1 and mTORC2 complexes may lead to the further suppression of stem cell survival and proliferation [80,129]. While the previously introduced small molecule inhibitors PP242 and Torin 1 did not show clinical benefits in trial, Torin 2 showed some clinical success [79,92,93,96,97]. Our recent findings have demonstrated that Torin 2 was more effective in suppressing stem cell growth and self-renewal as demonstrated by reduced neurosphere sizes [94,95,130]. In fact, while Torin 1 and XL388 delayed the self-renewal of GBM stem cells, Torin 2 completely halted stem cell self-renewal [130]. With these findings, mTOR inhibition emerges as a promising strategy to target CSCs in the treatment of GBM. A recent study utilizing Pim-1, another S/T kinase, demonstrated its potential influence in the regulation of GBM stem cells as its inhibition led to the eradication of stem-like neurosphere cells, and this effect can interact with Akt/mTOR to control the size and cell viability of neural stem cell neurospheres [131]. Future investigation is required to further define the role of the PI3K/Akt/mTOR pathway in stem cell quiescence and evaluate its significance in GBM treatment.

## 5. Conclusions

While substantial progress has been made in understanding the mechanisms of the mTOR complexes since their discovery nearly three decades ago, numerous molecular and cellular properties of mTOR are yet to be discerned, particularly the non-canonical regulation of these complexes. Furthermore, the regulation and activation of mTORC2 are still being investigated. As the deregulation of mTOR is seen in multiple tumor types, the mTOR pathway represents a promising therapeutic target. Allosteric inhibitors of mTOR, rapamycin, and its derivatives incompletely inhibit mTORC1 and result in the activation of mitogenic pathways. Although “second-generation” ATP-competitive compounds have shown promise in inhibiting both mTORC1 and mTORC2, a third-generation inhibitor of mTOR, RapaLink-1, effectively targets multiple domains in both complexes. The success of these compounds in targeting GBM CSC and treating GBM remains to be further investigated.

## Figures and Tables

**Figure 1 cells-13-00409-f001:**
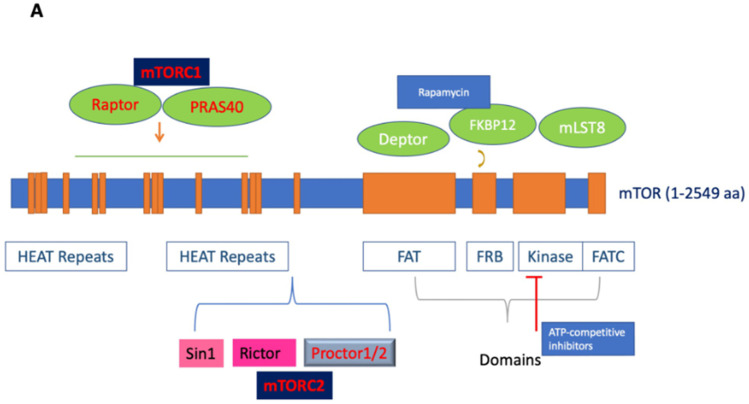
(**A**). Figure depicting the mTOR structure, consisting of 2549 amino acids, which has several domains essential for the activity of mTORC1 and mTORC2. mTORC1 has several binding partners, including Raptor and PRAS40, while mTORC2 has Rictor, Sin1, and Protor1/2. In addition, the figure shows FKBP12, a rapamycin-binding partner. See text for additional details. (**B**). Illustration of the mTORC1 and mTORC2 signaling pathway. Activated PI3K phosphorylates phosphatidylinositol 4,5-bisphosphate (PIP2) to form PIP3. PIP3 binds to PDK1/AKT via the PH-domains to mediate Akt phosphorylation, which is further facilitated by the activation of mTORC2. Activated Akt promotes both the phosphorylation of PRAS40 on Thr246 and the inhibition of TSC1/TSC2 complex activity, resulting in increased GTP-bound Rheb levels and mTORC1 activation. Activated mTORC1 then acts on multiple protein substrates, including 4E-BP1, S6K, and PRAS40. The phosphorylation of 4E-BP1 and S6K by activated mTORC1 regulates multiple functions, including mRNA translation, cellular growth, and proliferation. Furthermore, S6K provides negative feedback to inhibit the insulin-signaling pathway via IRS, which is disinhibited along with subsequent mitogenic pathways following the prolonged inhibition of mTORC1 (dotted line). Representative inhibitors: rapamycin and rapalogues; ATP binding inhibitors; Akt, PI3K, and dual PI3K/mTOR inhibitors are presented (for details, see Table 1 and text).

**Table 1 cells-13-00409-t001:** Clinical trials of PI3K/Akt/mTOR signaling inhibitors in glioblastoma.

Drug	Target	CTN(s)
Rapamycin (Sirolimus)	mTOR	NCT00047073 (completed, phase I/II), 13 GBM patients
Temsirolimus (Torisel)	mTOR	NCT00016328 (completed, phase II), 33 GBM patients; NCT00022724 (completed, phase 1/II), 49 malignant glioma patients
Everolimus	mTOR	NCT00387400 (completed, phase I), 32 GBM patients treated with temozolomide + everolimus; NCT00515086 (early termination due to slow enrollment and protocol-defined stopping rule), 41 GBM patients
AZD8055	ATP-competitive inhibitor of mTOR	NCT01316809 (completed, phase I), 22 glioma patients
AZD2014	ATP-competitive inhibitor of mTOR	NCT02619864 (completed, phase I), 15 GBM patients
Perifosine	Akt inhibitor	NCT02238496 (completed, phase I), 10 GBM patients treated with perifosine and temsirolimus; NCT01051557 (completed, phase I/II), 36 glioma patients treated with perifosine and temsirolimus
CC-115	mTOR/DNA-PK dual inhibitor	NCT01353625 (completed, phase I), 14 GBM patients
Paxalisib (GDC-0084)	mTOR/PI3K dual inhibitor	NCT03522298 (active, not recruiting, phase II), 32 newly diagnosed GBM patients
Samotolisib (LY3023414)	mTOR/PI3K dual inhibitor	NCT03213678 (active, not recruiting, phase II) for recurrent GBM; NCT03155620 (recruiting, phase II) for recurrent or refractory GBM
Buparlisib (BKM-120)	Pan-PI3K inhibitor	NCT01339052 (completed, phase II), 65 GBM patients
RMC-5552	Selective Bi-Steric inhibitor of mTORC1	NCT05557292 (active, not recruiting, phase I), 48 GBM patients
Fimepinostat (CUDC-907)	PI3K/HDAC dual inhibitor	NCT03893487 (active, not recruiting), 30 diffuse intrinsic pontine glioma, recurrent anaplastic astrocytoma, GBM, glioma, or medulloblastoma patients

Abbreviations: Clinical Trials Network, CTN; mechanistic target of rapamycin, mTOR; National Clinical Trial, NCT; glioblastoma, GBM; adenosine triphosphate, ATP; DNA-dependent protein kinase, DNA-PK; phosphoinositide 3-kinase, PI3K; mTOR complex 1, mTORC1; and histone deacetylase, HDAC.

## Data Availability

The original contributions presented in the study are included in the article/Appendix A, further inquiries can be directed to the corresponding author/s.

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
