# Peer review of "Discrete Mechanistic Target of Rapamycin Signaling Pathways, Stem Cells, and Therapeutic Targets"

_cells, 2024, doi:10.3390/cells13050409_

Round 1
Reviewer 1 Report
Comments and Suggestions for Authors
This review describes clinical trials and summary of the roles of mTOR in cancer. These descriptions are up to date and helpful. The review also aims to summarise the basics of mTOR regulation, with more limited success. In several instances, it seems to focus on relatively older literature before there were structures of the complexes available. It would have been helpful if the authors tried to reconcile older studies with more recent ones that include cell-free reconstitution and structural approaches to define regulatory mechanism. It is surprising that the manuscript does not even mention cancer-associated mTOR mutations, and it claims that cancer-associated Akt mutations do not exist. mTORC1 is regulated by both growth factors and nutrition. The nutritional regulation is barely mentioned in the text. It should be described, or there should be an explicit statement that they are omitting this regulation, with a reference to a nutrition-oriented review. There are several careless mistakes in the figures and text. I have tried to summarise these points in the following.
1. Fig. 1A. mTOR1 should be mTORC1
2. Fig. 1A. Daptor should be Deptor
3. Fig 1B. Proctor-1 should be Protor
4. Fig. 1B. “PIP 2” shoule be PIP3
5. Fig. 1B. What is the difference between black and purple arrows in the mTORC2 outputs?
6. Fig. 1. Legend says “mTORC1 has several binding partners, including Raptor, PRAS40, and HEAT repeats.” HEAT repeats are not binding partners. They are part of the mTOR.
7. Fig. 1B. There is a floating grey line under the Raptor” box. Not clear what it means.
8. In the mTORC2 complex, it looks as though there are two copies of mSIN1. The entire complex is a dimer, but only one copy is shown for the other subunits. It is not clear why there are two copies of mSIN1.
9. Lines 81-82 “…which is further facilitated by activation of mTORC2.” There are two mechanisms of mTORC2 activation that have been proposed: PH domain of mSIN1 binding to PIP3 and activation of mTORC2 by Akt. The review has no reference to either and is unclear as to the activation mechanism.
10. Lines 82-84, seems to be saying that both PRAS40 phosphorylation and TSC2 phosphorylation result in increased GTP-Gheb. This is not the case. They inhibit by completely different mechanisms.
11. Sentences on lines 93-95 are not clear.
12. Line 100 refers to something called mTORC. Should be “mTORC1 and mTORC2” or mTOR complexes. However, this makes no sense because Rapamycin inhibits only mTORC1. The sentence should be: “The mechanism by which rapamycin inhibits mTORC1 is via formation of a heterodimer with the small 12 kDa FKBP12, which then directly binds and potently inhibits mTORC1 [20,21].
13. Line 102 says that both mTORC1 and mTORC2 are activated by GTP-Rheb. This is not true. Rheb activaes only mTORC1.
14. Line 104, mTOR should be mTORC1 or mTOR in mTORC1.
15. Line 118 mTOR should be mTORC2
16. Lines 125-126. Reference 39 has nothing to do with Sin1 mutants. There is a problem with the references.
17. Line 136. This is an old reference. A more recent study showed that DEPTOR interacts with mTOR by using both its PDZ domain and its tandem DEP domains (PMID 34519268)
18. Lines 147-148 states: “While mutations in PI3K and mTOR have been detected, no mutations in Akt have been found.” There are at several well characterised mutants of Akt1 associated with cancer and therapy resistance. While Akt1 mutations are less common that PIK3CA, the statement in the text is not correct.
19. Line 179. mTOR should be mTORC1
20. Lines 194-195: The Rapalink links an FRB-binding moiety to an ATP-binding site moiety. The manuscript says Rapalink consists of “a FRB domain linked to a TORKi”, which is clearly nonsense.
21. Line 213. The authrs should state exactly what they mean be a “partial inhibitor”.
Author Response
Reviewer Response: We remain grateful to reviewer for their constrictive comments and we have address all of them.
Reviewer #1:
- “Figure 1B would benefit from an update…” – Figure 1B is now corrected with clearer labels, and now includes inhibitors of Akt, PI#K and dual inhibitor of PI3K/mTOR
- “I would also suggest adding a few more examples of mTOR targeting in other types of cancer…” – While the focus of our review remains on GBM, we have now included a supplementary table presenting the clinical trials ofmTOR inhibitors in different cancer types (lines 225-231).
- “The role of mTORC2 in glioblastoma stem cells. This should be described more” – We have elaborated further on the preclinical trials examining mTORC1/2 inhibitors on neurosphere formation and targeting of GBM stem cells (lines 403-435).
- “Minor typos” – We reviewed and addressed any remaining typos.
Reviewer 2 Report
Comments and Suggestions for Authors
Here, the authors present a review of the mTOR signaling pathway, its role in stem cells and cancer, and its potential as a therapeutic target for glioblastoma.
Major points:
The article covers a wide range of topics related to mTOR signaling, from its molecular mechanisms and regulation to its implications for various biological processes and diseases.
The article provides a comprehensive overview of the current state of knowledge and research on mTOR and its complexes, highlighting the challenges and opportunities for future studies.
The article discusses the latest developments and innovations in the field of mTOR inhibition, such as the discovery of novel compounds and strategies to overcome resistance and toxicity in cancers, with a specific focus on glioblastoma.
Minor points:
Figure 1B would benefit from an update, as it is quite confusing as is. Several inhibitors are missing from the scheme, such as NVP BEZ235 (Dactolisib) that was tested in several clinical trials.
I would also suggest adding a few more examples of mTOR targeting in other types of cancer where it proved to be beneficial, such as:
A phase III trial of everolimus, an mTORC1 inhibitor, in combination with exemestane, a steroidal aromatase inhibitor, in patients with hormone receptor-positive, HER2-negative advanced breast cancer showed a significant improvement in median progression-free survival compared to placebo plus exemestane (10.6 vs. 4.1 months).
A phase II trial of temsirolimus, an mTORC1 inhibitor, in patients with advanced renal cell carcinoma showed an overall survival benefit of 10.9 months compared to 7.3 months for interferon-alpha, a standard therapy.
A phase II trial of AZD2014, a dual mTORC1/2 inhibitor, in patients with recurrent endometrial cancer showed an overall response rate of 14%, with a median progression-free survival of 4.6 months.
This would help justify the relevance of targeting mTOR in cancers.
Finally, the article does not provide enough evidence or data to support some of the claims or hypotheses, such as the role of mTORC2 in glioblastoma stem cells. This should be described more.
Comments on the Quality of English Language
minor typos
Author Response
Reviewer Response: We remain grateful to the reviewers for their constructive comments and we have addressed all of them
Reviewer #2:
- “Many excellent recently published review articles…” – We have updated references to be more current, including a 2021 mTOR review by Popova (Reference #10) and a 2023 mTOR review by Panwar et al. (Reference #3).
- “Preclinical and clinical studies on targeting mTOR in glioblastoma stem cells” – While there are no clinical studies specifically examining GBM stem cells, we have elaborated further on the preclinical trials, as mentioned in Reviewer #1 response (see above).
- “There are several mistakes in Figure 1A…” – Figure 1A has been corrected.
- “PDCD-4 should be PDCD4” – Corrected (line 91)
- “Two S6 kinases (S6Ks) that act downstream of mTORC1 are S6K1 and S6K2…” – Nomenclature has been corrected (lines 63-64).
- “The statement ‘no mutations in Akt have been found’ is incorrect.” – We have clarified this statement to now reads like, “Although mutations in Akt are rare, ample mutations in upstream effectors of Akt, such as PTEN and PI3K, have been identified” (lines 146-147).
Reviewer 3 Report
Comments and Suggestions for Authors
The review article entitled “Discrete mTOR Signaling Pathways, Stem Cells and Therapeutic Targets” by Jhanwar-Uniyal et al. is clearly presented but does not include recent information. Most of the references cited are old. It included a long introduction focused on the regulation of the mTOR signaling pathway and some information is also repeated in later sections. Many excellent recently published review articles on this topic are much more informative and they could have been cited. The review should have elaborated on recent articles on preclinical and clinical studies on targeting mTOR in glioblastoma stem cells.
Comments:
There are several mistakes in Figure 1A. mTOR1 and mTOR2 should be mTORC1 and mTORC2. Raptor in the pink box should be Rictor. Daptor should be Deptor. There should not be any space between PRAS and 40.
Line 92: PDCD-4 should be PDCD4.
Some of the information provided is misleading or incorrect. For example, it was mentioned that mTORC1 promotes protein translation via activation of p70 S6 kinase (S6K)….(line 1-62) and S6K exists as two distinct kinases, p90 S6K and p70/p85 S6K (line 63-64). Two S6 kinases (S6Ks) that act downstream of mTORC1 are S6K1 and S6K2 whereas p90 ribosomal S6 kinase or RSK is activated by the MAPK pathway.
Line 147-148: The statement “no mutations in Akt have been found” is incorrect.
Author Response
Reviewer Response: We remain grateful to reviewer for their constrictive comments and we have address all of them.
Reviewer #1:
- “Figure 1B would benefit from an update…” – Figure 1B is now corrected with clearer labels, and now includes inhibitors of Akt, PI#K and dual inhibitor of PI3K/mTOR
- “I would also suggest adding a few more examples of mTOR targeting in other types of cancer…” – While the focus of our review remains on GBM, we have now included a supplementary table presenting the clinical trials ofmTOR inhibitors in different cancer types (lines 225-231).
- “The role of mTORC2 in glioblastoma stem cells. This should be described more” – We have elaborated further on the preclinical trials examining mTORC1/2 inhibitors on neurosphere formation and targeting of GBM stem cells (lines 403-435).
- “Minor typos” – We reviewed and addressed any remaining typos.
Reviewer #2 and 3:
- “Many excellent recently published review articles…” – We have updated references to be more current, including a 2021 mTOR review by Popova (Reference #10) and a 2023 mTOR review by Panwar et al. (Reference #3).
- “Preclinical and clinical studies on targeting mTOR in glioblastoma stem cells” – While there are no clinical studies specifically examining GBM stem cells, we have elaborated further on the preclinical trials, as mentioned in Reviewer #1 response (see above).
- “There are several mistakes in Figure 1A…” – Figure 1A has been corrected.
- “PDCD-4 should be PDCD4” – Corrected (line 91)
- “Two S6 kinases (S6Ks) that act downstream of mTORC1 are S6K1 and S6K2…” – Nomenclature has been corrected (lines 63-64).
- “The statement ‘no mutations in Akt have been found’ is incorrect.” – We have clarified this statement to now reads like, “Although mutations in Akt are rare, ample mutations in upstream effectors of Akt, such as PTEN and PI3K, have been identified” (lines 146-147).